# Thermoelectric active cooling for transient hot spots in microprocessors

Yihan Liu[1], Hao-Yuan Cheng[2], Jonathan A. Malen [2,3] ✉ & Feng Xiong [1] ✉

Modern microprocessor performance is limited by local hot spots induced at high frequency by busy integrated circuit elements such as the clock generator. Locally embedded thermoelectric devices (TEDs) are proposed to perform active cooling whereby thermoelectric effects enhance passive cooling by the Fourier law in removing heat from the hot spot to colder regions. To mitigate transient heating events and improve temperature stability, we propose a novel analytical solution that describes the temperature response of a periodically heated hot spot that is actively cooled by a TED driven electrically at the same frequency. The analytical solution that we present is validated by experimental data from frequency domain thermal reflectance (FDTR) measurements made directly on an actively cooled Si thermoelectric device where the pump laser replicates the transient hot spot. We herein demonstrate a practical method to actively cancel the transient temperature variations on circuit elements with TEDs. This result opens a new path to optimize the design of cooling systems for transient localized hot spots in integrated circuits.

Over the years, the size of transistors in silicon chips has steadily decreased due to Moore's law, resulting in increased heat generation per unit volume. As a consequence, the electronics industry has been faced with the challenge of dissipating higher heat fluxes to maintain a low operating temperature, since overheating can cause reduced circuit performance, increased leakage power, or even breakdown of transistors[1,2]. Circuits such as clock generators and arithmetic and logic units (ALU) create high-frequency heat fluxes of $1 \, kW \, cm^{-2}$ on average, and peak heat fluxes five times that of the surrounding areas create localized hot spots on the microprocessor (Fig. 1a)[3,4]. Therefore, cooling systems at the chip size are over designed and a more targeted strategy is necessary to dissipate high frequency thermal fluxes at local hot spots[5]. In this case, embedded TEDs are attractive solutions since they offer site-specific localized cooling that can be actively controlled by the application of electrical current.

Recent research has focused on removing the heat generated at hot spots by integrating thermoelectric refrigerators based on

classic thermoelectric materials such as $Bi_2Te_3$ that have high thermoelectric figures of merit, $ZT = S^2T/\rho\kappa$, of around 1 at room temperature where $S$ is the Seebeck coefficient, $\rho$ is the electrical resistivity, and $\kappa$ is the thermal conductivity[6–12]. High ZT materials with low $\kappa$ are optimal for thermoelectric refrigeration whereby heat is transferred from cold to hot since low $\kappa$ limits the amount of heat driven back from hot to cold by temperature gradients. However, high ZT materials are inferior for active cooling of hot spots where thermoelectric effects work in parallel to heat diffusion and thus enhance the passive cooling from hot to cold (Fig. 1b). Active cooling with high ZT materials is thus less effective than passive cooling with materials such as copper that possess two orders of magnitude larger $\kappa$[13].

Prior research has investigated the transient response of thermoelectric refrigeration devices, but not for the purpose of active cooling. Particular emphasis was given to the transient supercooling effect, which occurs when a single or continuous train of current pulses, as opposed to constant current, is driven through a

[1]Department of Electrical and Computer Engineering, University of Pittsburgh, Pittsburgh, PA, USA. [2]Department of Mechanical Engineering, Carnegie Mellon University, Pittsburgh, PA, USA. [3]Department of Materials Science and Engineering, Carnegie Mellon University, Pittsburgh, PA, USA. ✉e-mail: jonmalen@andrew.cmu.edu; f.xiong@pitt.edu

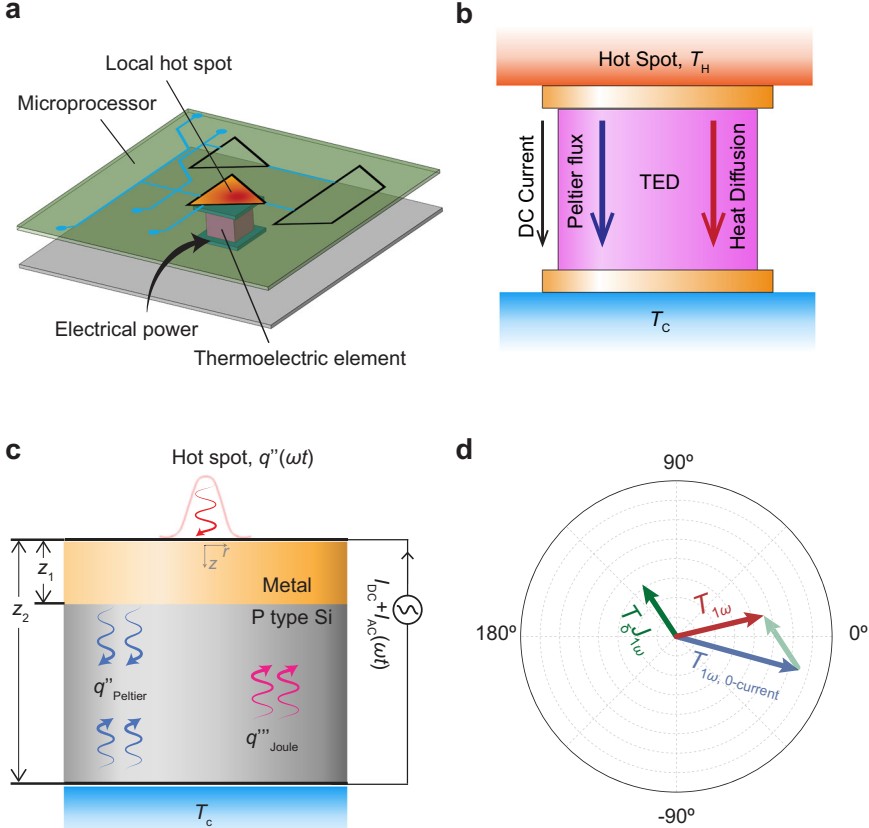

**Fig. 1 | Integrated TED for in-chip transient thermal management. a** An embedded TED under the localized hot spot generated by a high frequency circuit within a microprocessor. **b** Illustration of active cooling. In contrast to a conventional thermoelectric refrigerator, heat is removed from the hot side to the cold side through a combination of Peltier cooling and natural heat diffusion. Consequently, high $\kappa$ materials are selected to facilitate more significant natural heat conduction. **c** Schematic of the 3D model of a TED with transient thermoelectric active cooling. The top metal coated surface is heated by a radially gaussian sinusoidal heat source. Sinusoidal AC current is driven vertically through the device with the same frequency as the heat source. **d** Illustration of model's analytical result demonstrates the final hot spot's first harmonic temperature profile. This profile results from the combination of passive cooling and thermoelectric cooling, with the thermoelectric cooling effect being influenced solely by the electrical current's parameters and the operating frequency.

thermoelectric refrigerator. When compared to applying optimal DC current to minimize the steady-state temperature at the cold-side, a current pulse can temporarily induce a greater temperature difference between the hot and cold-sides if the hot-side is maintained at room temperature[5,14–22]. These papers demonstrate that instantaneous Peltier cooling at the cold side of the device can further depress temperatures before it is opposed by parasitic Joule heat, that is generated volumetrically and takes time to diffuse. These findings motivate our research direction and identify open questions in transient thermoelectric active cooling that still need to be addressed before its technological adoption is realistic. Prior research on supercooling employs electric pulses with durations typically spanning milliseconds or more[5,14,23–27]. These timescales are much longer than the timescale typically encountered in the transient operation of high frequency circuits in microprocessors. There thus exists a scarcity of experimental and theoretical explorations of transient thermoelectric active cooling at short timescales. Secondly, instead of the spatially uniform and temporally constant heat flux at the hot spot considered in the majority of supercooling studies, a more accurate representation of a hot spot within a microprocessor involves a transient excursion characterized by a spatially distributed heat flux[28]. Lastly, there is no theoretical research guiding the design of AC waveforms for achieving optimal thermoelectric cooling of hot spot transients described by arbitrary waveforms.

In this paper, we develop a model for analyzing the transient thermal response of a hot spot actively cooled by an AC-powered TED. Specifically, we consider a device that is locally surface-heated by a high frequency sinusoidal source with a radially Gaussian distribution. Our model presents the analytical solution and suggests a method for tuning the AC electrical input to arbitrarily modulate the phase and amplitude of the first harmonic temperature response at the hot spot. This approach can be used to stabilize and, in some cases, completely cancel transient temperature excursions. To validate our model results, we fabricate a micro-pillar TED from highly p-doped silicon. We select p-doped Si for its compatibility with silicon chips, high $\kappa$ for passive cooling performance, as well as its ability to generate high heat flux Peltier cooling without inducing significant Joule heating, owing to its high Seebeck coefficient and low electrical resistivity[29,30]. We measure the TED's thermal behavior with frequency-domain thermoreflectance (FDTR). FDTR employs a modulated pump laser to periodically heat the TED, and a probe laser to measure the resultant temperature response through variations in the surface reflectance of the TED[31,32]. Active cooling is achieved by applying a frequency-matched sinusoidal AC current parallel to the primary thermal gradient. Under the guidance of our model, we can completely negate hot spot temperature first harmonic oscillations with amplitudes up to 2.35 K on the surface of TED. Our validated model therein provides valuable insight to optimize thermoelectric active cooling of micro-TEDs for transient thermal management in microprocessors.

## Results and discussion

### Modeling a TED subjected to transient thermoelectric active cooling

We consider a 3D planar TED subjected to periodic surface heating from a radially Gaussian heat source at an angular frequency of $\omega$. In order to facilitate thermoelectric cooling, frequency-matched AC current complemented by a DC offset is applied (Fig. 1c).

Our analysis begins with the heat diffusion equation for the thermoelectric material layer with volumetric Joule heating, which is expressed as follows:

$$\frac{1}{\alpha_{TE}}\frac{\partial T_{TE}}{\partial t} = \nabla^2 T_{TE} + J^2 \rho_{TE}/\kappa_{TE} \qquad (1)$$

where $T_{TE}(r, z, t)$ is the temperature profile of the thermoelectric (TE) material layer, $\alpha_{TE}$ represents its thermal diffusivity, $J$ is current density, $t$ signifies time, and $r$ and $z$ denote the spatial coordinates, as depicted in Fig. 1c. The applied sinusoidal AC current is uniformly distributed across the metal contact, rendering $J$ independent of spatial position. We decompose the current and temperature profiles into two components: the steady-state component ($0\omega$) and the first order harmonic component ($1\omega$) represented mathematically as $J = J_{0\omega} + J_{1\omega}\sin(\omega t)$ and $T = T_{0\omega}(r, z) + T_{1\omega}(r, z)\sin(\omega t)$, respectively. Second and higher harmonic temperature profiles are excluded in the process of solving the heat equation, as they do not have a significant impact on the solution, as validated by our experiments. A Fourier transformation is applied to Eq. (1) to obtain the following two heat equations for the $0\omega$ and $1\omega$ components.

$$0 = \nabla^2 T_{TE0\omega} + (J_{0\omega}^2 + J_{1\omega}^2/2)\rho_{TE}/\kappa_{TE}$$
$$\frac{i\omega}{\alpha_{TE}}T_{TE1\omega} = \nabla^2 T_{TE1\omega} + 2J_{0\omega}J_{1\omega}\rho_{TE}/\kappa_{TE} \qquad (2)$$

The effects of the radially Gaussian, periodic heat source and Peltier effect induced by the AC current are manifested in the boundary conditions:

$$-\kappa_m \frac{\partial T_m}{\partial z} = \frac{2P}{\pi w_0^2}\exp\left(-\frac{2r^2}{w_0^2}\right)$$
$$-\kappa_m \frac{\partial T_m}{\partial z} = -\kappa_{TE}\frac{\partial T_{TE}}{\partial z} + S_{TE}\left(J_{0\omega} + J_{1\omega}\exp(i\omega t)\right)\left(T_{TE0\omega} + T_{TE1\omega}\exp(i\omega t)\right) \qquad (3)$$

where the subscript m designates metal, $P$ represents the power of heat source, and $w_0$ stands for the $1/e^2$ spot radius of the radially Gaussian source. Notably, the Peltier term at the interface between the metal and thermoelectric layers induces transient active cooling.

Despite the general challenges of obtaining an analytical solution for a 3D heat equation, the radial symmetry of our model simplifies the mathematical difficulty and allows us to derive an analytical solution for this 3D heat equation. With the Hankel transformation and numerical approximations, we successfully derived an analytical solution (Fig. 1d) for the first harmonic temperature profile in Eq. (2) for p-Si TED (refer to Supplementary Note 1 for the detailed derivation):

$$T_{1\omega} = T_\delta(J_{0\omega}, \omega) \cdot J_{1\omega} + T_{1\omega 0} \qquad (4)$$

where $T_{1\omega 0}$ stands for the measured first harmonic temperature in the absence of applied current (i.e., the result of passive cooling). The active cooling component, $T_\delta(J_{0\omega}, \omega)J_{1\omega}$, is exclusively determined by the parameters of the applied current, irrespective of the amplitude of the heat source. Hence, by maintaining other parameters constant, it is possible to arbitrarily tune the effect of active cooling through adjustments to the amplitude and phase of the AC current.

Our subsequent analysis and experiments are focused to scenarios where DC current is set to 0 ($J_{0\omega} = 0$), and only AC current is

applied, thereby eliminating the influence of conventional steady-state thermoelectric cooling. There exists a trade-off between transient active cooling capability and steady-state temperature. It is possible to reduce the steady-state temperature with limited positive DC offset applied ($J_{0\omega} > 0$), but it will concomitantly diminish the efficacy of active cooling. Conversely, applying a negative DC offset ($J_{0\omega} < 0$) can lead to an increase in the steady-state temperature; however, the transient active cooling performance is increased as well (see Supplementary Note 2 for more details on the effect of DC current on transient active cooling performance).

### Validating the model result with FDTR measurement

To validate the analytical solution presented above, we engineered a p-type silicon micro-TED. Our micro-TED, fabricated from highly doped p-Si, consists of a 100 μm high Si pillar. Heating will be applied to a metal contact atop the pillar, and cold-side metal contacts symmetrically flank the bottom of the pillar (Fig. 2a, c). To reduce the contact resistance between the Au electrodes and p-Si, we introduce a layer of Ni between the Au layer and p-Si, creating an Ohmic contact. In order to eliminate any potential paths for unintended electrical current flow, the hot-side exclusively establishes electrical contact with a 30 μm × 30 μm window of p-Si, with the remaining area being insulated by a 100 nm-thick SiO$_2$ layer. To minimize current crowding at the edges of the heated contact, current pathways are lengthened and thus equalized by maintaining a 50 μm lateral separation between the edges of the heated contact and cold-side contact.

FDTR, described in detail elsewhere[32], employs a 488 nm continuous wave (CW) laser (referred to as the pump laser) to induce heating in the sample. Simultaneously, a 532 nm CW laser (referred to as the probe laser) is used to measure temperature via the thermoreflectance of Au. The pump laser is directed through an electro-optic modulator (EOM) to sinusoidally modulate its power. In order to conduct the transient active cooling measurement, we apply AC current through the micro-TED using an arbitrary function generator. It simultaneously generates a reference signal for the EOM and Lock-in amplifier ensuring frequency and phase synchronization between the AC current and heating. The arbitrary function generator also provides us with the capability to adjust the phase of the AC current in relation to the hot spot heating (Fig. 2b).

Under zero current conditions, corresponding to the $T_{1\omega 0}$ in Equation (4), we extract p-Si thermal conductivity and the metal-Si thermal boundary conductance through using them as fitting parameters to align the analytical prediction to the phase vs. frequency data. The Si thermal conductivity is 120 ± 7 W m$^{-1}$ K$^{-1}$, which is 96% ± 5.6% of the reported values in reference[33]. The thermal boundary conductance between Si and Ni is 60 ± 17 MW m$^{-2}$ K$^{-1}$, which is far less than the value reported in reference, 348 MW m$^{-2}$ K$^{-1}$ [34]. We hypothesize that the Ni silicide layer formed at the interface during the rapid thermal annealing reduces the interface thermal conductance (see Supplementary Note 3).

During the transient active cooling measurement, accomplished by varying the amplitude of the applied AC current ranging up to 90 mA, we observe a frequency-dependent positive phase shift in comparison to the conventional FDTR data (depicted in Fig. 2d). Phase values indicate that temperature leads heating by as much as 7° when the frequency is 100 kHz, with the AC amplitude set at 90 mA. Conventional FDTR theory, which yields negative phase values for this material system, fails to provide an explanation for this phenomenon since the temperature response should consistently lag behind the heat source[31]. However, we can explain this observation with our analytical prediction by including thermoelectric effects. According to the numerical value of Equation (4), phase of $T_\delta$ falls within the range of 110° to 130° when the frequency is below 1 MHz. Hence, when we apply a sinusoidal AC current in phase with the heat input, the resultant $T_\delta(J_{0\omega}, \omega) \cdot J_{1\omega}$ causes a positive phase shift in $T_{1\omega}$. This results in a net

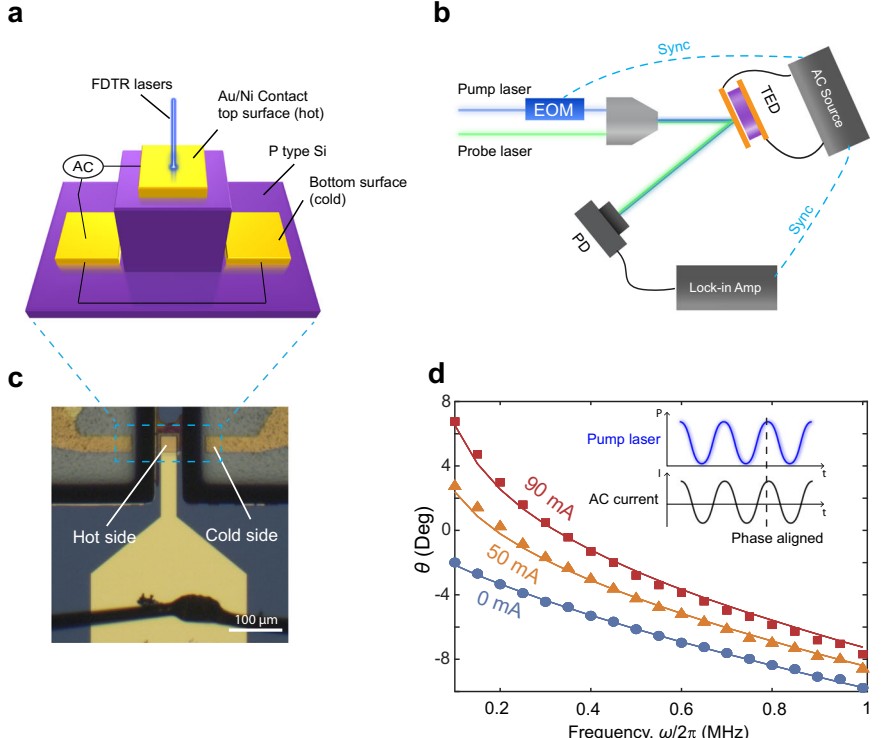

**Fig. 2 | p-Si micro TED sample and transient active cooling measurement.**
**a** Schematic of p-Si micro TED. **b** FDTR measurement setup and transient active cooling measurement setup with AC source included. **c** Optical image of p-Si micro TED. **d** Measured (points) and theoretical prediction (solid line) of phase-frequency curve in conventional FDTR measurement (when AC = 0) and transient active cooling measurement. Heat input and AC current are phase aligned.

positive phase value if the amplitude of the AC current is sufficiently high (as illustrated in Fig. 1d). The model prediction (represented by the solid lines in Fig. 2d) agree with the experimentally measured values over the entire frequency range. This validates the accuracy of our model and our decision to neglect higher harmonic contributions to the temperature profiles.

The disparity between the measured phase from the numerical value at frequencies higher than 500 kHz can be attributed to the presence of parasitic inductance stemming from the bonding wires and the geometric design of Au/Ni contact. Given the exceptionally low electrical resistance of the p-Si micro-TED, this inductance introduces phase delay in the AC current, particularly at high frequency. We have leveraged this discrepancy to estimate the magnitude of the parasitic inductance, allowing us to quantify the extent to which it introduces phase distortion to the AC signal at a particular frequency (see Supplementary Note 4).

**First harmonic temperature cancellation**
Through Equation (4), we have discovered that we can completely cancel the first harmonic temperature ($T_{1\omega} = 0$) if we can generate a transient thermoelectric cooling component with the same magnitude but opposite phase to $T_{1\omega0}$, i.e., $T_\delta (J_{0\omega}, \omega) J_{1\omega} = -T_{1\omega0}$ (Fig. 3a). The first harmonic temperature canceling phenomenon ensures that only a steady-state heat input will contribute to the temperature gradient on the surface and holds promise for reducing or completely canceling hot spots due to transient heat loads. As explained in the previous section, the impact of transient thermoelectric cooling is directly linked to $J_{1\omega}$. As a result, we can control the phase and amplitude of the transient thermoelectric cooling component by adjusting both the phase and amplitude of AC current. To make the transient thermoelectric cooling component out of phase with $T_{1\omega0}$, we need to apply an additional phase, which is $180° + \phi(T_{1\omega0}) - \phi(T_\delta)$, to $J_{1\omega}$. To match the amplitude of the transient thermoelectric

cooling component with $T_{1\omega0}$, the required AC current density defined by Equation (4) is $|J_{1\omega}| = |T_{1\omega0} / T_\delta|$.

Experiments were conducted to cancel the first harmonic temperature with a 10 kHz heat input and opposing phase on the AC current. Analysis of the temperature vs. AC amplitude data obtained from the transient active cooling measurements reveals a linear reduction in the amplitude of $T_{1\omega}$ as the AC amplitude increases (Fig. 3b). This reduction rate remains consistent even for varying laser heating powers, providing evidence that $T_\delta$ is independent of the heat input amplitude at the hot spot, as confirmed by the analytical findings. The maximum $T_{1\omega0}$ that can be canceled at a frequency of 10 kHz for our setup was 2.35 ± 0.24 K, due to the limited load capacity of our AC power supply. This corresponded to an absorbed laser power of 836 μW which is equivalent to a periodic heat flux of 18.5 kW cm⁻² in average amplitude, which is one order of magnitude larger than the heat flux generated by a hot spot in a microprocessor[3].

The analytical solution also proves that as long as a high enough AC current is supplied, it is possible to effectively stabilize and cancel the transient temperature regardless of the heat source amplitude. While the AC current reduces the amplitude of $T_{1\omega}$, the additional electrical energy input will result in Joule heating that increases the second order harmonic temperature and the steady-state temperature (i.e., $T_{0\omega}$). In principle it is also possible mitigate or eliminate the higher harmonic heating effects by superimposing higher harmonic AC currents. We estimate the increase in $T_{0\omega}$ resulting from AC Joule heating using $\frac{|J_{1\omega}|^2 \rho_{Si}(z_2-z_1)^2}{4\kappa_{Si}}$, as shown by the solid line in Fig. 3b. The transient peak temperature is a balance of the two effects that depends on the magnitude of $J_{1\omega}$ and the operating frequency.

In Fig. 3c, we present the phase-AC amplitude curve on polar coordinate. It is evident that $T_{1\omega}$ undergoes a phase flip as $J_{1\omega}$ is increased. Initially $\phi(T_{1\omega})$ is aligned with $\phi(T_{1\omega0})$, then it shifts to 180° - $\phi(T_{1\omega0})$ after cancellation. However, instead of the sudden phase flip

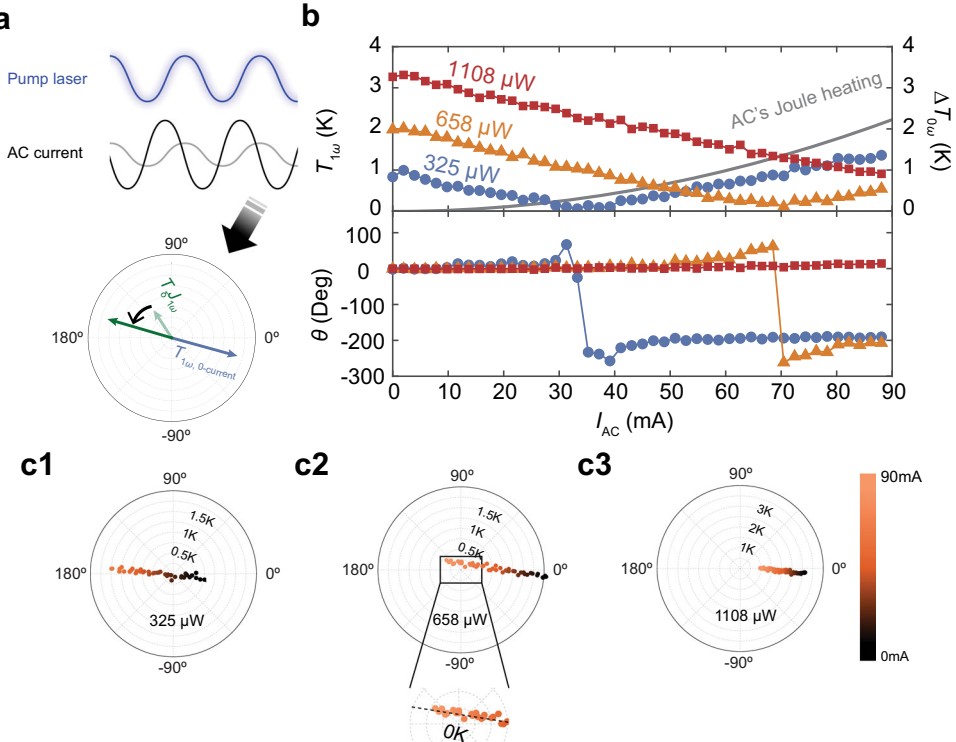

**Fig. 3 | First harmonic temperature cancellation. a** Schematic of strategy for canceling $T_{1\omega}$ through adjustment of the applied AC current. To make the thermoelectric cooling component oppose the passive cooling component, additional phase, specified by the analytical solution is imposed on the AC current. **b** The measured amplitude reduction and cancellation of $T_{1\omega}$, estimation of steady-state temperature rise due to Joule heating (upper panel), and phase value of $T_{1\omega}$ (lower panel) across various absorbed laser powers. Once $T_{1\omega}$ reaches zero, subsequent increases to AC current will lead to a 180° flip in its phase. **c1–3** Polar coordinate representation of **b**. The zoomed in view in **c2** shows that the misalignment between thermoelectric cooling component (dashed line extrapolate from data) and $T_{1\omega0}$ contributes to the gradual phase transition, since $T_{1\omega}$ does not precisely reach zero.

predicted by our model, we observe phase fluctuations around the temperature cancellation point and a gradual phase transition from $\phi(T_{1\omega0})$ to nearly $180° - \phi(T_{1\omega0})$. There are two reasons for the fluctuations: 1). Around the temperature cancellation point, the measured $T_{1\omega}$ amplitude will be close to or lower than the noise floor of our measurement setup, leading to noise fluctuations in the measured phase. 2). When $T_\delta (J_{0\omega}, \omega) J_{1\omega}$ does not strictly oppose $T_{1\omega0}$, the resultant $T_{1\omega}$ does not precisely pass through the zero point as AC amplitude increases. Hence, the measured phase undergoes a gradual transition instead of a sudden flip (see lower panel of Fig. 3c2).

### Metric of transient active cooling performance

Similar to the coefficient of performance (COP) in conventional thermoelectric elements, we can also formulate the coefficient of performance for transient active cooling (COP$_{trans}$) at the transient temperature cancellation point. This metric is computed as the temporal root mean square value (RMS) of heat flux at the hot spot divided by the input power areal density at the transient temperature cancellation point. Hence higher COP$_{trans}$ means less power is required to reach the same level of transient active cooling. Since our heat source is sinusoidal in time, we can estimate its heat flux's RMS as $q''_{RMS} = \frac{q''_{peak}}{\sqrt{2}} = \frac{1}{\sqrt{2}\pi w_0^2} P_{laser}$. The expression for COP$_{trans}$ can be deduced as $\frac{\sqrt{2}\left|T_\delta(J_{0\omega}, \omega)\right|^2 P_{laser}}{|T_{1\omega0}|^2 \rho_{Si} h_{TED} \pi w_0^2}$ (refer to Supplementary Note 5 for details of derivation). In Fig. 4a, we use contours to show how COP$_{trans}$ depends on $q''_{RMS}$ and frequency for highly p-doped Si. COP$_{trans}$ will experience a decrease when $T_{1\omega0}$ increases from increased $q''_{RMS}$. The transport properties of highly p-doped Si allow it to induce a large amplitude of $T_\delta$ (high $S$) with a low $T_{1\omega0}$ (high κ) without generating substantial Joule

heating (low ρ). This leads to a high COP$_{trans}$ for the p-Si micro-TED. For example, to cancel the transient effects of a sinusoidal hot spot operating at a frequency of 10 kHz with an RMS heat flux of 10 kW cm$^{-2}$, the necessary AC power per unit area is just 1.01 kW cm$^{-2}$.

Operating frequency is a pivotal parameter that dictates the value of COP$_{trans}$. Notably, when the frequency is sufficiently high ($f \gg \frac{\alpha_{TE}}{2\pi z_2^2}$, which is ~13.2 kHz for our sample), COP$_{trans}$ decreases exponentially with increasing square root value of operating frequency (see Supplementary Note 5). These results are further supported by the experimental data we have obtained in the frequency range from 10 kHz to 1 MHz (Fig. 4b).

Having successfully compared our COP$_{trans}$ data with theory for our specific device, in Fig. 4c we compare the theoretical values of COP$_{trans}$ between active cooling devices made from p-Si and p-Bi$_2$Te$_3$. The COP$_{trans}$ of p-Si is higher than p-Bi$_2$Te$_3$ because it has a far higher thermal conductivity (120 W m$^{-1}$ K$^{-1}$ vs. 1.84 W m$^{-1}$ K$^{-1}$)[35]. Supplementary Note 6 provides other assumed properties of p-Bi$_2$Te$_3$ and additional details about COP$_{trans}$ of p-Si and p-Bi$_2$Te$_3$. Notably, as the thickness of the metal electrode on the hot side decreases, the ratio of COP$_{trans}$ between p-Si and p-Bi$_2$Te$_3$ devices increases further. In the limiting scenario where the thickness of the metal electrode is 0 nm, the COP$_{trans}$ of p-Si devices is two orders of magnitude greater than that of p-Bi$_2$Te$_3$ devices. This implies that to mitigate the transient temperature response generated by the same heat source, p-Bi$_2$Te$_3$ devices would require 100 times more electrical power than p-Si devices. When the thickness of the metal electrode increases this disparity is reduced because the electrode itself contributes to passive cooling. These findings highlight the importance of material selection and device design for effective active cooling.

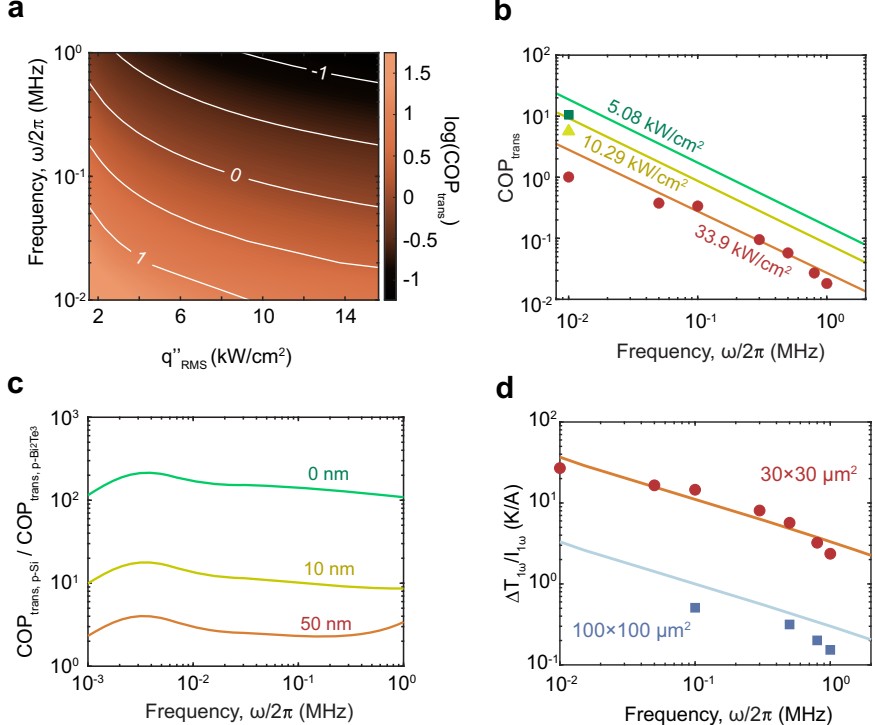

**Fig. 4 | COP$_{trans}$ and improvement of energy efficiency by sizing TED's hot-side contact area. a** Numerical COP$_{trans}$ value at transient temperature cancellation point determined from the analytical solution. **b** Measured COP$_{trans}$ vs frequency for various absorbed heat flux amplitudes. Both experimental and theoretical data demonstrate that an increase in the amplitude of the heat input, along with an increase in the operating frequency, results in a decrease in COP$_{trans}$. **c** Theoretical ratio of COP$_{trans}$ between p-Si and p-Bi$_2$Te$_3$ devices as a function of frequency for

q''$_{RMS}$ = 10 kW cm$^{-2}$ and hot side metal electrode thicknesses of 0 nm, 10 nm and 50 nm. Reduced electrode thickness magnifies the contrast of passive cooling effects between these two materials thereby increase the ratio. **d** $T_{1\omega}$'s change when unit AC current applied measured on TED with different hot-side contact areas. For smaller contacts, the application of a unit AC current leads to a more significant reduction in $T_{1\omega}$. Consequently, sizing the hot-side contact to match the heat source can improve TED's power efficiency in transient active cooling.

To enhance the power efficiency of micro-TED it is optimal to match the hot-side contact area with the heat source area because active cooling performance depends solely on the current density, instead of the total applied current. To validate this observation, we measured the $T_{1\omega}$ reduction (defined as $\Delta T_{1\omega} = T_{1\omega} - T_{1\omega 0}$) per unit applied AC current on two p-Si micro-TED samples with hot-side contact areas of 30 × 30 μm$^2$ and 100 × 100 μm$^2$ respectively (Fig. 4d). The $T_{1\omega}$ reduction with the smaller contact is 10–20 times larger than that with the larger contact, which is comparable to the ratio between the two samples' contact areas as well as our analytical prediction. The measured $T_{1\omega}$ reduction for the larger contact area sample is smaller than the value predicted by our analytical result. This discrepancy results from current crowding because the path traveled by the current from the center of the contact (coinciding with our transient active cooling measurement spot) is significantly longer than the path from the edges of the contact. Consequently, current tends to congregate at the edge of the contacts resulting in a considerably lower current density at center of the contact. This phenomenon, in turn, leads to a reduction in the measured performance of transient active cooling. However, for samples with smaller contact areas, the path length to all points on the contact is similar and current crowding effects are weak, so the measured $T_{1\omega}$ aligns better with the prediction.

In summary, we provide a novel analytical solution for a thermoelectric active cooling device that is heated by a temporally sinusoidal and spatially gaussian hot spot, and concurrently cooled by an AC current with the same frequency. Our solution predicts the phase and amplitude of the temperature response on the device's surface and outlines clear strategies for modifying the temperature by tuning the AC current input only. To experimentally validate our solution, we fabricated a micro-thermoelectric device from p-type Si. We used

FDTR to mimic a periodically heated hot spot and measured the resultant temperature response. We find that our analytical solution successfully explains the observed phase shift and can be used to identify the phase shift in AC current needed to cancel the temperature variations. We achieve active cancellation of the hot spot temperature up to an amplitude of 2.35 K at a heat flux amplitude of 18.5 kW cm$^{-2}$ and frequency of 10 kHz under the guidance of our modeling result. Evaluation of active cooling performance indicates high power efficiency can be achieved on Si-based TEDs with a synergistic combination of significant passive cooling and active cooling when the hot side of the TED is comparable in size to the hot spot. More temporally complicated transient heat inputs, like those seen in real microprocessors can be decomposed into a summation of sinusoidal waves by Fourier transformation. Our analytical solution provides a starting point for stabilizing or completely canceling the hot spot temperature response and optimizing active cooling by TEDs for transient thermal management of microprocessors operating at high frequency.

## Methods
### Sample fabrication
Substrates consist of thermal SiO$_2$ of 100 nm thickness, grown on high p-doped silicon (the measured values of Seebeck coefficient and electrical resistivity of the p-doped silicon are 350 ± 10 μV K$^{-1}$ and 1.05 × 10$^{-3}$ Ω cm.). The areas of the cold-side and hot-side are defined by photolithography using SC-1827 as photoresist. Then the sample is immersed in a buffered oxide etch solution (7:1 BOE) for 30 seconds to etch the SiO$_2$ layer on the hot-side and cold-side. The silicon pillar is fabricated by an inductively coupled plasma reactive ion etch (ICP-RIE) using AZ-4620 with a thickness of 20 μm as photoresist dry etching mask. Metal contacts on the hot-side and cold-side are defined by

e-beam lithography using PMMA 950k A4 resist. 50 nm Ni is deposited after 1 min of Ar ion cleaning by e-beam evaporation. To form an ohmic contact between Ni and p-Si, a 90 second rapid thermal anneal (RTA) is conducted at 600 °C in forming gas (5% $H_2$, 95% Ar). Areas of gold wire and contact covering are defined by e-beam lithography followed by 150 nm Au deposition using e-beam evaporation. The sample is then wire bonded to a chip carrier using Al-Si bonding wire.

## FDTR setup

A 488 nm continuous wave (CW) laser (pump laser) is utilized to heat the sample, and a 532 nm CW laser (probe laser) is utilized to measure temperature via the thermoreflectance of Au. The pump laser is passed through an electro-optic modulator to sinusoidally modulate its power. Both lasers have a Gaussian-shaped beam profiles and the effective $1/e^2$ radius of the laser spot was $1.6 \pm 0.05$ µm after focusing by a 50× objective.

## Data availability

All data needed to reproduce figures are available on Figshare https://doi.org/10.6084/m9.figshare.25602021.

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

## Acknowledgements

This material is in part based upon work supported by Intel Corporation. In addition, this work was supported in part by the U.S. National Science Foundation under Grants ECCS-1901864, ECCS-1943683, ECCS-1901972, and the Army Research Office Award W911NF2310260.

## Author contributions

Y.L., J.A.M., and F.X. conceived the idea. Y.L. performed the modeling, calculation, and micro-TED fabrication. Y.L. and H.-Y.C. designed and worked on the FDTR measurements and transient active cooling measurements. The manuscript was written by Y.L. with input from all authors. All authors contributed to the analysis and discussion of the results leading to the manuscript.

## Competing interests

The authors declare no competing interests.
