## [Peer Review File · Nature Communications]

Thermoelectric Active Cooling for Transient Hot Spots in MicroprocessorsREVIEWER COMMENTS

Reviewer #1 (Remarks to the Author):

Thermoelectric have been considered as the most promising method to eliminate the hotspot of chips for twenty years. This work proposed an analytical model that describes temperature response of a periodically heated hot spot that is actively cooled by a TED driven electrically at the same frequency, and verified their model by frequency domain thermal reflectance (FDTR) measurements. I do think this idea of applying frequency-matched AC current to suppressed hot spot temperature is very innovative, which takes a substantial step towards application of thermoelectric transient effect. However, this work leaves much to be desired, and several key issues should be addressed.

1. This work only examines the frequency matching between the thermoelectric transient action and the simulated heat source, but the thermoelectric parameter of p-Si is not mentioned, and how the thermoelectric properties determine the final hot spot temperature has not been investigated. This problem may be the one of the most critical in determining what role thermoelectric transient effects can play in chip thermal management.
2. Currently, only AC part is considered to evaluate the temperature suppression and COP, how about the result after including the DC part?
3. According to eq. 14 and Fig. 4A, at fixed frequency and specified phase of AC current (which makes the active cooling component oppose the passive cooling), the phase of response shall be constant when only increasing the AC current amplitude. Which caused the variations of the phase in the low panel of Fig. 4B (especially for the case of 658 μ W absorbed laser power), despite the phase flipping?
4. Why 10 kHz is chosen to verify the cancelation of temperature variation amplitude? What about the cooling effect at higher frequencies in the range of conventional FDTR, like beyond 100 kHz?
5. Is it possible to quantify the phase lag of the caused parasitic inductance at high frequency and take it into consideration when tuning the AC input?
6. In line 252-255, the author claimed that larger Joule heating will increase the time-steady temperature and that the transient temperature reduction is balanced by the increased time-steady temperature. In the following section, they concluded that both increase frequency and source heat flux will reduce COP_{trans}. The question is, is there any specific point of frequency or source heat flux, beyond which the balance will be broken and the active cooling by TEDs is not worthwhile?
7. There is a lot of work to be done in writing and expression,

Reviewer #2 (Remarks to the Author):

The paper by Liu et al. "Thermoelectric Active Cooling for Transient Hot Spots in Microprocessors" reports an effective strategy of active cooling for transient hot spots in microprocessors by locally embedded thermoelectric devices fabricated from highly p-doped silicon. The authors established a model for analyzing the transient thermal response of a hot spot actively cooled by an AC-powered TED. Based on the model, they reported the transient temperature excursions can be canceled by tuning the AC electrical input. This is interesting; however, this manuscript is not well written and many concerns are not discussed. Therefore, I do not suggest accepting this current form.

1. The most important part of this article is the established theoretical model for active cooling by AC-powered TED to cancel transient temperature excursions. This is proven effective by canceling the hot spot temperature with an amplitude of 2.35 K at a heat flux amplitude of 18.5 kW/cm² and frequency of 10 kHz. However, it is hard to identify a sizeable novelty to be published at Nature Communications, because the mechanism of using Peltier cooling has been discussed in previous studies, i.e., the supercooling effect.

2. Similar to the first point. In my humble opinion, the statement “As yet, there are not analytical or experimental studies to model thermoelectric active cooling under a transient heat flux” is hard to justify. There have been certain studies modeling thermoelectric cooler and transient supercooling effects. The extension to the condition of “a transient heat flux” only requires adding a boundary condition.

3. The authors emphasized that high thermal conductivity is considered in this work compared with previous studies that frequently consider low thermal conductivity. However, this point is not discussed. To discuss this, reporting the temperature profile of the device may be helpful.

4. Please see Fig. 3(A). Is the sketch of the pump laser correct? The energy input by the pump laser should be positive.

5. Too many technical details are presented in the discussion, which confuses the manuscript. In my humble opinion, these details can be introduced together in a single section.

6. The sketches of phase and amplitude should be drawn on a polar coordinate instead of a rectangular coordinate, such as Fig. 3(B) and Fig. 4 (A), (C), and (D).

7. Some results need in-depth analysis. For example, the authors only discuss that the hot spot temperature can be canceled by Fig. 4(B) to support their strategy. However, why will θ experience a fluctuation when $T1\omega$ is close to zero?

8. It is expected to guide the active cooling process by the established model. For example, can the theoretical model predict the critical IAC for $T1\omega=0$ at different transient heat inputs? This should be discussed.

9. It is worth noting that the model seems not accurate when the hot end area is relatively large (100×100 μm^2) as shown in Fig. 6. This also needs further discussion.

Reply to Referee and Revisions Made

Nature Communications Manuscript #: NCOMMS-23-39295

Title: “Active Cooling for Transient Hot Spots in Microprocessors”

Authors: Yihan Liu, Hao-Yuan Cheng, Jonathan A. Malen, Feng Xiong

We appreciate the insightful comments and feedback regarding our submission referenced above. We have carefully considered all the comments and addressed them with point-by-point responses below, as well as through changes to the text of the manuscript. These comments requested additional acknowledgement of thermoelectric supercooling research, sought clarification of how active cooling performance changes with material properties and operating parameters, and inquired about discrepancies between our predictions and experiments. Each comment led to changes in the text that significantly improve the quality and clarity of our manuscript without changing its major conclusions. The reviewer comments are reproduced verbatim in *blue italics* below, followed by our responses and revisions made accordingly. The changes in the re-submitted manuscript itself have been **highlighted**. We are also providing a clean version of the re-submitted manuscript where the highlights are removed.

Reviewer #1

Thermoelectric have been considered as the most promising method to eliminate the hotspot of chips for twenty years. This work proposed an analytical model that describes temperature response of a periodically heated hot spot that is actively cooled by a TED driven electrically at the same frequency, and verified their model by frequency domain thermal reflectance (FDTR) measurements. I do think this idea of applying frequency-matched AC current to suppressed hot spot temperature is very innovative, which takes a substantial step towards application of thermoelectric transient effect. However, this work leaves much to be desired, and several key issues should be addressed.

We appreciate the reviewer’s positive comment: “*I do think this idea of applying frequency-matched AC current to suppressed hot spot temperature is very innovative, which takes a substantial step towards application of thermoelectric transient effect.*”.

1. This work only examines the frequency matching between the thermoelectric transient action and the simulated heat source, but the thermoelectric parameter of p-Si is not mentioned, and how the thermoelectric properties determine the final hot spot temperature has not been investigated. This problem may be the one of the most critical in determining what role thermoelectric transient effects can play in chip thermal management.

We agree that the thermoelectric parameters of p-Si should be reported. To clarify the values of the Seebeck coefficient and electrical resistivity of the p-Si we used to fabricate our sample, we added the measured values in the manuscript, section “Methods”, line 355-356:

The measured values of Seebeck coefficient and electrical resistivity of the p-doped silicon are $350 \mu\text{V K}^{-1}$ and $1 \times 10^{-3} \Omega \cdot \text{cm}$.

Not only the material’s thermoelectric properties but also its thermal conductivity will determine the transient **active cooling performance** and the final hot spot temperature. In active cooling, heat is pumped from hot spot to the cold end in the same direction as natural heat diffusion (see Fig. 1b). Thus materials with higher thermal conductivity will exhibit **better passive cooling performance**, which means lower $T_{1\omega_0}$ amplitude. In terms of thermoelectric cooling, highly doped Si can generate **significant Peltier effects** at metal/Si interface due to its high Seebeck coefficient ($350 \mu\text{V K}^{-1}$ for our sample). Furthermore, its low electrical resistivity ensures **low Joule heating** while current flowing. Our analysis on $\text{COP}_{\text{trans}}$ at the transient temperature cancellation point also confirms that both low $T_{1\omega_0}$ and high T_δ (i.e. strong transient thermoelectric modulation capability) will increase the TED’s efficiency in using energy. In general, to achieve lower transient hot spot temperature with limited electrical energy applied, a material should have higher Seebeck coefficient, lower electrical resistivity (for stronger thermoelectric modulation capability), and higher thermal conductivity (for better passive cooling). Consequently, highly doped silicon is a promising material for energy efficient transient active cooling.

To mitigate the confusion about the nature of thermoelectric active cooling, we have added a figure (Fig. 1b) to emphasize the alignment between natural heat diffusion and Peltier effects.

Figure 1b. Schematic of active cooling. In active cooling, thermoelectric effects drive a heat flux in parallel with natural heat diffusion and is thus additive to passive cooling.

To clarify the reason for selecting high p-doped silicon as the thermoelectric material, we add the following text in section “Introduction”, line 79-82:

We select p-doped Si for its compatibility with silicon chips, high κ for passive cooling performance, as well as its ability to generate high heat flux Peltier cooling without inducing significant Joule heating, owing to its high Seebeck coefficient and low electrical resistivity^{1,2}.

We emphasize contribution of p-Si’s thermoelectric properties to COP_{trans} in section “Results and discussion”, line 279-284:

The transport properties of highly p-doped Si allow it to induce a large amplitude of T_{δ} (high S) with a low $T_{1\omega 0}$ (high κ) without generating substantial Joule heating (low ρ). This leads to a high COP_{trans} for the p-Si micro-TED. For example, to cancel the transient effects of a sinusoidal hot spot operating at a frequency of 10 kHz with an RMS heat flux of 10 kW cm^{-2} , the necessary AC power per unit area is just 1.01 kW cm^{-2} .

We also add the following text of comparison between the theoretical COP_{trans} of p-Si and p-Bi₂Te₃ active cooling devices to emphasize how high thermal conductivity improve transient active cooling’s efficiency in section “Results and discussion”, line 290-302:

Having successfully compared our COP_{trans} data with theory for our specific device, in Fig. 4c we compare the theoretical values of COP_{trans} between active cooling devices made from p-Si and p-Bi₂Te₃. The COP_{trans} of p-Si is higher than p-Bi₂Te₃ because it has a far higher thermal conductivity ($120 \text{ W m}^{-1} \text{ K}^{-1}$ vs. $1.84 \text{ W m}^{-1} \text{ K}^{-1}$)³. Supplementary Note 6 provides other assumed properties of p-Bi₂Te₃ and additional details about COP_{trans} of p-Si and p-Bi₂Te₃. Notably, as the

thickness of the metal electrode on the hot side decreases, the ratio of COP_{trans} between p-Si and p-Bi₂Te₃ devices increases further. In the limiting scenario where the thickness of the metal electrode is 0 nm, the COP_{trans} of p-Si devices is two orders of magnitude greater than that of p-Bi₂Te₃ devices. This implies that to mitigate the transient temperature response generated by the same heat source, p-Bi₂Te₃ devices would require 100 times more electrical power than p-Si devices. When the thickness of the metal electrode increases this disparity is reduced because the electrode itself contributes to passive cooling. These findings highlight the importance of material selection and device design for effective active cooling.

Figure 4c. Theoretical ratio of COP_{trans} between p-Si and p-Bi₂Te₃ devices as a function of frequency for $q''_{RMS} = 10 \text{ kW/cm}^2$ and hot side metal electrode thicknesses of 0 nm, 10 nm and 50 nm. Reduced electrode thickness magnifies the contrast of passive cooling effects between these two materials thereby increase the ratio.

2. Currently, only AC part is considered to evaluate the temperature suppression and COP, how about the result after including the DC part?

Just as the reviewer mentioned, in our analysis and experiments, the DC offset is fixed at 0 in order to avoid the influence of steady-state thermoelectric cooling generated by DC currents on transient active cooling. To clarify the DC level of the AC current we applied in analysis and experiment, we add the following text in manuscript, section “Results and discussion”, line 144-146:

Our subsequent analysis and experiments are focused to scenarios where DC current is set to 0 ($J_{0\omega} = 0$), and only AC current is applied, thereby eliminating the influence of conventional steady-state thermoelectric cooling.

According to our model's $T_{1\omega}$ analytical result (Equation (4)), only T_{δ} will be affected by DC offset. We have added a supplementary note to specifically discuss the dependence of T_{δ} 's amplitude on DC current (Supplementary Information, line 110-132).

Supplementary Note 2: Effect of DC Current on Transient Active Cooling Performance

Equation (S14) signifies that T_{δ} is a function of the DC current. To investigate this correlation, we conduct experiments wherein a DC current is superimposed onto the 10 kHz AC current utilized for transient temperature cancellation measurement. Additionally, we conduct a comparative analysis between the measured T_{δ} 's amplitude and the theoretical predictions derived from Equation (S14) (as depicted in Fig. S2). It is worth noting that the theoretical values exhibit a turning point at around 120 mA. We hypothesize that this turning point is linked to the optimal DC current in steady-state Peltier cooling. When the DC current falls below this optimal current, an increase in the DC current leads to a reduction in $T_{0\omega}$, consequently diminishing the transient Peltier heat flux since transient Peltier flux is $S_{Si}(T_{0\omega}J_{1\omega} + T_{1\omega}J_{0\omega})$. This, in turn, results in a decline in the performance of transient active cooling. Conversely, when the DC current surpasses the optimal current, an increase in the DC current raises $T_{0\omega}$, thereby amplifying the effectiveness of transient active cooling. Owing to constraints related to the bonding wire in our sample and the power budget of the arbitrary function generator, we encountered difficulties in applying a DC offset at the anticipated turning point position. Nonetheless, our experimental results confirmed the trend of T_{δ} within the experimentally accessible range of DC current. Consequently, we find that when applying transient active cooling with limited DC supply, there is a trade-off between the transient active cooling capability and steady-state temperature. It is possible to elevate transient active cooling performance by introducing a negative DC offset onto the AC current; however, this concurrently results in an increased steady-state temperature. Conversely, applying a positive DC offset can lead to a reduction in steady-state temperature but concomitantly diminishes the efficacy of active cooling.

Fig. S2. Dependence of T_{δ} 's amplitude on DC offset at 10 kHz. Measured value (points) aligns with theoretical prediction (solid line) to the left of turning point (~ 120 mA).

3. According to eq. 14 and Fig. 4A, at fixed frequency and specified phase of AC current (which makes the active cooling component oppose the passive cooling), the phase of response shall be constant when only increasing the AC current amplitude. Which caused the variations of the phase in the low panel of Fig. 4B (especially for the case of $658 \mu\text{W}$ absorbed laser power), despite the phase flipping?

We thank the reviewer for this question. We observe phase fluctuations around the transient temperature cancellation point. Two reasons account for this phenomenon: 1). When the measured **temperature is close to or lower than the noise floor of our measurement setup**, the measured phase is not reliable. That causes arbitrary jumps of phase superimposed upon the anticipated phase flipping predicted by our model (around $I_{AC} = 35\text{mA}$ in measurement of $325 \mu\text{W}$ in Fig. 3b). 2). Continuous phase fluctuations and transitions instead of sudden flipping at the $T_{1\omega}$ elimination point are observed in the measurement of $658 \mu\text{W}$ in Fig. 3b. This is due to a **slight difference** between the $T_{1\omega}$ elimination phase (which oppose active cooling and passive cooling) and the actual phase we applied on the AC current. When the AC amplitude is close to the $T_{1\omega}$ elimination point (i.e. $|T_{\delta}J_{1\omega}|$ is close to $|T_{1\omega 0}|$), this misalignment will bring about $\theta(T_{1\omega})$'s gradual transition from $\theta(T_{1\omega 0})$ to $\theta(T_{1\omega 0})+180^{\circ}$, instead of a sudden flipping.

To clarify the phase fluctuation around $T_{1\omega}$ cancellation point, we add the following text in the manuscript, line 247-256:

In Fig. 3c, we present the phase-AC amplitude curve on polar coordinate. It is evident that $T_{1\omega}$ undergoes a phase flip as $J_{1\omega}$ is increased. Initially $\phi(T_{1\omega})$ is aligned with $\phi(T_{1\omega 0})$, then it shifts to $180^\circ - \phi(T_{1\omega 0})$ after cancellation. However, instead of the sudden phase flip predicted by our model, we observe phase fluctuations around the temperature cancellation point and a gradual phase transition from $\phi(T_{1\omega 0})$ to nearly $180^\circ - \phi(T_{1\omega 0})$. There are two reasons for the fluctuations: 1). Around the temperature cancellation point, the measured $T_{1\omega}$ amplitude will be close to or lower than the noise floor of our measurement setup, leading to noise fluctuations in the measured phase. 2). When $T\delta(J_{0\omega}, \omega) J_{1\omega}$ does not strictly oppose $T_{1\omega 0}$, the resultant $T_{1\omega}$ does not precisely pass through the zero point as AC amplitude increases. Hence, the measured phase undergoes a gradual transition instead of a sudden flip (see lower panel of Fig. 3c2).

Figure 3c2. Polar representation of first harmonic temperature cancellation when absorbed heat input is $658 \mu\text{W}$. Lower panel in **c2** depicts the misalignment between thermoelectric cooling component (dashed line extrapolate from data) and $T_{1\omega 0}$ contributes to the gradual phase transition during the transient temperature cancellation process, since $T_{1\omega}$ cannot reach 0. We find that the phase misalignment is $4\text{-}5^\circ$

4. Why 10 kHz is chosen to verify the cancelation of temperature variation amplitude? What about the cooling effect at higher frequencies in the range of conventional FDTR, like beyond 100 kHz?

We thank the reviewer for pointing out that we only include the transient cancellation data at 10kHz in the manuscript. Our analytical result indicates that 13.2 kHz is a critical frequency beyond which COP_{trans} will decrease exponentially with square root of frequency. Hence 10 kHz is an operating frequency that showcases characteristics of transient thermoelectric cooling performance at high frequency while maintaining reasonable power efficiency. At higher frequencies (> 100 kHz), our ability to cancel transient heat flux is restricted by our limited AC power supply and the low COP_{trans} . Consequently, for higher frequencies (e.g. 1 MHz) we can cancel only a very small heat flux, which results in significant noise in the measured data. Considering these factors, we believe that 10 kHz is sufficient to effectively illustrate transient active cooling. Transient temperature cancellation data measured at higher frequency is added in Supplementary Note 7 in the Supplementary Information, line 220-228:

Supplementary Note 7: Transient Temperature Cancellation at High Frequency (> 100 kHz)

Figure S6 illustrates the transient temperature cancellation observed at higher operating frequencies. As the operating frequency increases, the heat flux that can be cancelled is reduced for the same AC amplitude. This phenomenon stems from the COP_{trans} decreasing with the increased frequency at high frequency.

Figure S6. Transient temperature cancellation observed under higher frequency (>100 kHz)

We add the following text to emphasize the boundary of high-frequency regime derived from our model results, in section “Results and discussion”, line 285-289:

Operating frequency is a pivotal parameter that dictates the value of COP_{trans} . Notably, when the frequency is sufficiently high ($f \gg \frac{\alpha_{TE}}{2\pi z_2^2}$, which is ~ 13.2 kHz for our sample), COP_{trans} decreases exponentially with increasing square root value of operating frequency (see Supplementary Note 5). These results are further supported by the experimental data we have obtained in the frequency range from 10 kHz to 1 MHz (Fig. 4b).

5. Is it possible to quantify the phase lag of the caused parasitic inductance at high frequency and take it into consideration when tuning the AC input?

With the extracted L/R value from measured phase of T_δ ($\approx 0.25 \mu\text{H}/\Omega$ as shown in Fig. S3b), it is possible for us to determine the phase lag resulting from package's parasitic inductance. The total impedance of sample is $R+i2\pi fL$, thus the extra phase lag is $\arctan(2\pi fL/R) = \arctan(2\pi f \times 0.25 \times 10^{-6})$ for any specific frequency.

We clarify this point in Supplementary Note 4, line 155-158 in Supplementary Information:

We find the presence of parasitic inductance within the device package introduces an increasing phase delay as the frequency rises. Hence the phase distortion on T_δ resulting from the parasitic inductance is $\arctan(2\pi fL/R)$, where L is the value of inductance and R is device's resistance.

6. In line 252-255, the author claimed that larger Joule heating will increase the time-steady temperature and that the transient temperature reduction is balanced by the increased time-steady temperature. In the following section, they concluded that both increase frequency and source heat flux will reduce COP_{trans} . The question is, is there any specific point of frequency or source heat flux, beyond which the balance will be broken and the active cooling by TEDs is not worthwhile?

The existence of such a specific point hinges on the application scenario. When transient active cooling is applied in order to reduce or cancel the effects of high frequency temperature variations and create a stable thermal environment, the steady-state temperature rise caused by Joule heating from the AC current does not impact the transient temperature cancellation and there will be no such point. However, when transient active cooling is applied to reduce the peak

temperature, there is a specific AC amplitude, beyond which it does more harm than good. This is because the transient temperature reduction is proportional to AC current amplitude according to our analytical result, but the steady-state temperature rise due to Joule heating is a parabolic function in terms of AC current amplitude. Hence the steady-state temperature increase will eventually become larger than transient active cooling at a specific AC current amplitude. As the operating frequency increases, the specific AC amplitude will reduce because T_δ drops while the steady-state temperature rise is independent of frequency.

To clarify the quadratic proportionality between steady-state temperature rise result and applied AC amplitude, and to provide readers with a better understanding of the steady-state temperature rise, the following text is added into manuscript within section “Results and discussion”, line 240-246:

While the AC current reduces the amplitude of $T_{1\omega}$, the additional electrical energy input will result in Joule heating that increases the second order harmonic temperature and the steady-state temperature (i.e. $T_{0\omega}$). In principle it is also possible mitigate or eliminate the higher harmonic heating effects by superimposing higher harmonic AC currents. We estimate the increase in $T_{0\omega}$ resulting from AC Joule heating using $\frac{|J_{1\omega}|^2 \rho_{Si} (z_2 - z_1)^2}{4\kappa_{Si}}$, as shown by the solid line in Fig. 3b. The transient peak temperature is a balance of the two effects that depends on the magnitude of $J_{1\omega}$ and the operating frequency.

Figure 3b The measured amplitude reduction and cancellation of $T_{1\omega}$, estimation of steady-state temperature rise due to Joule heating (upper panel), and phase value of $T_{1\omega}$ (lower panel) across various absorbed laser powers.

7. There is a lot of work to be done in writing and expression

To improve clarity and coherence, we carefully revised the manuscript and divided our results and discussion into several subsections. We also refined our language and structure to better convey our ideas.

Reviewer #2

The paper by Liu et al. "Thermoelectric Active Cooling for Transient Hot Spots in Microprocessors" reports an effective strategy of active cooling for transient hot spots in microprocessors by locally embedded thermoelectric devices fabricated from highly p-doped silicon. The authors established a model for analyzing the transient thermal response of a hot spot actively cooled by an AC-powered TED. Based on the model, they reported the transient temperature excursions can be canceled by tuning the AC electrical input. This is interesting; however, this manuscript is not well written and many concerns are not discussed. Therefore, I do not suggest accepting this current form.

We sincerely thank the reviewer for the feedback. In this revision we have carefully refined our language and restructured the manuscript to better convey our ideas.

1. The most important part of this article is the established theoretical model for active cooling by AC-powered TED to cancel transient temperature excursions. This is proven effective by canceling the hot spot temperature with an amplitude of 2.35 K at a heat flux amplitude of 18.5 kW/cm² and frequency of 10 kHz. However, it is hard to identify a sizeable novelty to be published at Nature Communications, because the mechanism of using Peltier cooling has been discussed in previous studies, i.e., the supercooling effect.

As the reviewer points out, the supercooling effect is a transient thermoelectric cooling phenomenon that uses Peltier cooling. According to some highly cited and recently published numerical⁴⁻⁹ and experimental¹⁰⁻¹³ studies about supercooling (all now cited by the paper), it describes the temporary temperature drop at the cold side of a **thermoelectric refrigerator** when subjected to a **pulsed electric current, typically with pulse width of >1 ms**, while being heated by a **temporally constant, spatially uniform heat flux**. The majority of these studies employ only **1-D model** analytical or numerical solutions.

There are additional problems to be resolved if we intend to apply transient thermoelectric cooling or supercooling to hot spot thermal management in microprocessors. 1). Within the context of microprocessor hot spot thermal management, heat is transferred from the hot spot to a cold side or reservoir. As a result, the conventional investigations focusing on the performance and efficiency of thermoelectric refrigeration aren't directly translatable to this scenario. 2). Typically, integrated circuits operate at frequencies varying from a few hundred kHz to GHz. More studies need to be conducted to explore the performance of transient thermoelectric cooling at high frequencies. 3). In real-world scenarios, the heat flux generated by hot spots in ICs is not constant; it exhibits spatial nonuniformity and temporal fluctuations¹⁴. Therefore, further research is needed to investigate the coupling between transient heat flux and thermoelectric cooling in such dynamic conditions. 4). We have observed that in most of these supercooling studies, the electrical driving signal for the TEC is a square pulse. While there are also other numerical simulation studies exploring and experimenting with the impact of different waveforms on supercooling effects (such as sawtooth, triangular or t^n waveforms), it remains challenging to derive waveform designs that can achieve optimal transient thermoelectric cooling effects from these numerical simulation studies.

To address transient thermal management of hot spots in microprocessors we herein report novel experiments and easily transferrable analytical modelling of a **3-D thermoelectric device, that incorporates a transient heat flux input and electrical current with a sinusoidal waveform**. Diverging from the conventional thermoelectric refrigerator, our focus centers on the unconventional use of the TED for **transient active cooling**. Here heat is transferred from the hot side (hot spot) to cold side (or cold reservoir). To emulate the hot spot in a chip that better matches real-world conditions, we consider a **radially Gaussian hot spot with frequencies ranging from 10 kHz to 1 MHz**. Furthermore, **the analytical results** derived from our model

describe the dependence of transient thermoelectric cooling performance on device's material properties (thermal conductivity, Seebeck coefficient, electrical conductivity, etc.) and the properties of the electrical current (AC amplitude, DC amplitude, frequency, etc.).

Our results define the necessary phase misalignment between the transient heat flux and AC input for achieving **the best transient active cooling performance**. The analytical result obtained from our model based on a sinusoidal waveform, holds significance for the analysis of transient active cooling in hot spots subjected to **transient heat fluxes with arbitrary waveforms**. In the experimental section, we fabricate a micro-TED, with both the hot end and cold end maintained at sizes in the tens of micrometers. As a result, the measurements we obtained are closer to the performance of a TED that could feasibly be integrated into ICs.

Hence, our study builds upon prior supercooling studies, but encompasses more realistic and general scenarios for in-chip thermal management.

To emphasize the unconventional thermoelectric cooling scenario we are considering, we add the following text in section "Introduction", line 46-48:

However, high ZT materials are inferior for active cooling of hot spots where thermoelectric effects work in parallel to heat diffusion and thus enhance the passive cooling from hot to cold (Fig. 1b).

To emphasize our focus on addressing the challenges associated with applying thermoelectric supercooling to hot spot thermal management in microprocessors, we add the following text in section "Introduction", line 59-71:

These findings motivate our research direction and identify open questions in transient thermoelectric active cooling that still need to be addressed before its technological adoption is realistic. Prior research on supercooling employs electric pulses with durations typically spanning milliseconds or more^{10-13,15-17}. These timescales are much longer than the timescale typically encountered in the transient operation of high frequency circuits in microprocessors. There thus exists a scarcity of experimental and theoretical explorations of transient thermoelectric active cooling at short timescales. Secondly, instead of the spatially uniform and temporally constant heat flux at the hot spot considered in the majority of supercooling studies, a more accurate representation of a hot spot within a microprocessor involves a transient excursion characterized by a spatially distributed heat flux¹⁴. Lastly, there is no theoretical research guiding

the design of AC waveforms for achieving optimal thermoelectric cooling of hot spot transients described by arbitrary waveforms.

2. Similar to the first point. In my humble opinion, the statement “As yet, there are not analytical or experimental studies to model thermoelectric active cooling under a transient heat flux” is hard to justify. There have been certain studies modeling thermoelectric cooler and transient supercooling effects. The extension to the condition of “a transient heat flux” only requires adding a boundary condition.

We agree with this reviewer that the effect of a transient heat flux in analyzing the heat equation is adding a boundary condition on the surface of the metal contact. We note that the application of these boundary conditions, and their validation through experiments, is designed to enable **more generalized and realistic scenarios** that distinguish our work from previous research on supercooling. As mentioned in our response to this reviewer’s first concern, both the radially Gaussian hot spot and the transient heat flux have heightened the complexity in reaching an analytical solution to the 3D heat equation with internal heat generation resulting from Joule heating. We obtained an analytical solution for this 3D heat equation and boundary conditions by employing radial symmetry, Fourier transformation, Hankel transformation, and through the validated assumption of neglecting higher order harmonic temperature responses (see Supplementary Note 1). This analytical result assists us in explaining the positive phase shift observed in FDTR measurements and provides guidance for optimizing the AC waveform to achieve complete cancellation of the first harmonic temperature response. Furthermore, we can extract the dependence of transient thermoelectric cooling performance at high frequency from this analytical result. We note that our study is not only how to solve a 3D heat equation with complicated boundary conditions but, more importantly, how our analytical results, based on these more realistic boundary conditions, help us optimize transient thermoelectric cooling across a wide range of heating events, device dimensions, and materials properties.

3. The authors emphasized that high thermal conductivity is considered in this work compared with previous studies that frequently consider low thermal conductivity. However, this point is not discussed. To discuss this, reporting the temperature profile of the device may be helpful.

We agree with the reviewer that our manuscript lacks discussion of the how material's high thermal conductivity improves the transient active cooling performance and efficiency. Different from conventional thermoelectric refrigerator, in active cooling, heat is conveyed from hot side to cold side so that the Peltier heat flux aligns with the direction of natural heat diffusion (as depicted in Fig. 1b). High thermal conductivity creates large natural heat diffusion from the hot end to the cold end and better passive cooling performance compared to low thermal conductivity materials. Thus, materials possessing not only high Seebeck coefficient but also high thermal conductivity are preferred.

To mitigate the confusion about thermoelectric active cooling (relative to thermoelectric refrigeration), we add a figure (Fig. 1b) to emphasize the alignment between natural heat diffusion and Peltier effect.

Figure 1b. Schematic of active cooling. In active cooling, thermoelectric heat flux is in parallel with natural heat diffusion to improve the performance of passive cooling.

We also revised the manuscript to emphasize how p-Si's high thermal conductivity improves its COP_{trans} in the section "Results and discussion", line 279-284:

The transport properties of highly p-doped Si allow it to induce a large amplitude of T_δ (high S) with a low $T_{1\omega 0}$ (high κ) without generating substantial Joule heating (low ρ). This leads to a high COP_{trans} for the p-Si micro-TED. For example, to cancel the transient effects of a sinusoidal hot spot operating at a frequency of 10 kHz with an RMS heat flux of 10 kW cm^{-2} , the necessary AC power per unit area is just 1.01 kW cm^{-2} .

We also add the following text of comparison between the theoretical $\text{COP}_{\text{trans}}$ of p-Si and p-Bi₂Te₃ active cooling devices to emphasize how high thermal conductivity improve transient active cooling's efficiency in section "Results and discussion", line 290-302:

Having successfully compared our $\text{COP}_{\text{trans}}$ data with theory for our specific device, in Fig. 4c we compare the theoretical values of $\text{COP}_{\text{trans}}$ between active cooling devices made from p-Si and p-Bi₂Te₃. The $\text{COP}_{\text{trans}}$ of p-Si is higher than p-Bi₂Te₃ because it has a far higher thermal conductivity ($120 \text{ W m}^{-1} \text{ K}^{-1}$ vs. $1.84 \text{ W m}^{-1} \text{ K}^{-1}$)³. Supplementary Note 6 provides other assumed properties of p-Bi₂Te₃ and additional details about $\text{COP}_{\text{trans}}$ of p-Si and p-Bi₂Te₃. Notably, as the thickness of the metal electrode on the hot side decreases, the ratio of $\text{COP}_{\text{trans}}$ between p-Si and p-Bi₂Te₃ devices increases further. In the limiting scenario where the thickness of the metal electrode is 0 nm, the $\text{COP}_{\text{trans}}$ of p-Si devices is two orders of magnitude greater than that of p-Bi₂Te₃ devices. This implies that to mitigate the transient temperature response generated by the same heat source, p-Bi₂Te₃ devices would require 100 times more electrical power than p-Si devices. When the thickness of the metal electrode increases this disparity is reduced because the electrode itself contributes to passive cooling. These findings highlight the importance of material selection and device design for effective active cooling.

Figure 4c. Theoretical ratio of $\text{COP}_{\text{trans}}$ between p-Si and p-Bi₂Te₃ devices as a function of frequency for $q''_{\text{RMS}} = 10 \text{ kW/cm}^2$ and hot side metal electrode thicknesses of 0 nm, 10 nm and 50 nm. Reduced electrode thickness magnifies the contrast of passive cooling effects between these two materials thereby increase the ratio.

4. Please see Fig. 3(A). Is the sketch of the pump laser correct? The energy input by the pump laser should be positive.

We thank the reviewer for bringing this up. We are sorry for the confusion here. The aim of inner sketch of pump laser and AC current's waveform is to highlight the phase alignment in the transient active cooling measurement. The pump laser's power should always be non-negative. To avoid any confusion on this sketch of pump laser and AC current's waveform comparison, we add P-t and I-t coordinate within the inner panel of Fig. 2d.

Figure 2d. Phase-Frequency data obtained from phase aligned transient active cooling measurement. Inner sketch of waveforms of pump laser and AC are revised accordingly.

5. Too many technical details are presented in the discussion, which confuses the manuscript. In my humble opinion, these details can be introduced together in a single section.

We thank the reviewer for this suggestion. Part of technical details about the sample fabrication process and the FDTR setup and measurements are now in “Methods” in the revised manuscript. Details of the analytical solution and the analysis of COP_{trans} are included in the Supplementary Information.

6. The sketches of phase and amplitude should be drawn on a polar coordinate instead of a rectangular coordinate, such as Fig. 3(B) and Fig. 4 (A), (C), and (D).

We thank the reviewer for pointing this out. We agree that incorporating a polar form of temperature can better assist readers in understanding our model's analytical results. Thus we revised Fig. 1d and Fig. 3a as shown below:

Figure 1d and 3a in polar form

7. Some results need in-depth analysis. For example, the authors only discuss that the hot spot temperature can be canceled by Fig. 4(B) to support their strategy. However, why will θ experience a fluctuation when $T_{1\omega}$ is close to zero?

We thank the reviewer for raising this concern, which is also pointed out by reviewer 1. We kindly ask the reviewer to refer to our answer to reviewer 1's third question.

8. It is expected to guide the active cooling process by the established model. For example, can the theoretical model predict the critical IAC for $T_{1\omega}=0$ at different transient heat inputs? This should be discussed.

We thank the reviewer for the comment. Based on the analytical result, the critical AC current density for $T_{1\omega}$ cancellation is predicted as $|J_{1\omega}| = |T_{1\omega 0} / T_{\delta}|$. We clarify this point in manuscript, section "Results and discussion", line 226-227:

To match the amplitude of the transient thermoelectric cooling component with $T_{1\omega 0}$, the required AC current density defined by Equation (4) is $|J_{1\omega}| = |T_{1\omega 0} / T_{\delta}|$.

From the phase of T_{δ} extracted from Equation (S14) and extra phase distortion induced by parasitic inductance (See Supplementary Note 4), we can derive the theoretical predicted phase we need to add in AC current.

9. It is worth noting that the model seems not accurate when the hot end area is relatively large ($100 \times 100 \mu\text{m}^2$) as shown in Fig. 6. This also needs further discussion.

This inaccuracy results because the larger sample experiences more pronounced current crowding effects. For the sample with a large hot side electrode, the current tends to concentrate more towards the electrode edges, resulting in a lower current density in the electrode center region (also the measured region) compared to the average current density (Fig. R1). Thus lower transient active cooling performance is measured compared to the model's result.

To emphasize the current crowding's effect on the performance of thermoelectric active cooling, we add the following text in manuscript, section "Results and discussion", line 310-318:

The measured $T_{1\omega}$ reduction for the larger contact area sample is smaller than the value predicted by our analytical result. This discrepancy results from current crowding because the path traveled by the current from the center of the contact (coinciding with our transient active cooling measurement spot) is significantly longer than the path from the edges of the contact. Consequently, current tends to congregate at the edge of the contacts resulting in a considerably lower current density at center of the contact. This phenomenon, in turn, leads to a reduction in the measured performance of transient active cooling. However, for samples with smaller contact areas, the path length to all points on the contact is similar and current crowding effects are weak, so the measured $T_{1\omega}$ aligns better with the prediction.

Figure R1. Current crowding in samples with various hot end contact areas. Schematic of half of the device with current applied. For the sample with the larger hot side electrode, current density at center, which is also the spot we conduct the thermal measurement, will be significantly lower than that at the edge due to the longer pathway for current. Hence the measured transient active cooling performance is lower than the prediction. However, for the sample with a smaller hot side electrode, the path lengths for all points on the electrode are similar, and the problem of current crowding is minimal.

References

- 1 Jaziri, N. *et al.* A comprehensive review of Thermoelectric Generators: Technologies and common applications. *Energy Reports* **6**, 264-287 (2020).
[https://doi.org:https://doi.org/10.1016/j.egy.2019.12.011](https://doi.org/https://doi.org/10.1016/j.egy.2019.12.011)
- 2 Dhawan, R. *et al.* Si_{0.97}Ge_{0.03} microelectronic thermoelectric generators with high power and voltage densities. *Nature Communications* **11**, 4362 (2020).
<https://doi.org:10.1038/s41467-020-18122-3>
- 3 Rowe, D. M. *CRC handbook of thermoelectrics*. (CRC press, 2018).
- 4 Lv, H., Wang, X.-D., Wang, T.-H. & Meng, J.-H. Optimal pulse current shape for transient supercooling of thermoelectric cooler. *Energy* **83**, 788-796 (2015).
- 5 Gupta, M. P., Min-hee, S. S., Mukhopadhyay, S. & Kumar, S. in *2010 12th IEEE Intersociety Conference on Thermal and Thermomechanical Phenomena in Electronic Systems*. 1-7 (IEEE).
- 6 Zhou, Q., Bian, Z. & Shakouri, A. Pulsed cooling of inhomogeneous thermoelectric materials. *Journal of Physics D: Applied Physics* **40**, 4376 (2007).
- 7 Manno, M., Wang, P. & Bar-Cohen, A. in *13th InterSociety Conference on Thermal and Thermomechanical Phenomena in Electronic Systems*. 413-420 (IEEE).
- 8 Asaadi, S., Khalilarya, S. & Jafarmadar, S. Numerical study on the thermal and electrical performance of an annular thermoelectric generator under pulsed heat power with different types of input functions. *Energy Conversion and Management* **167**, 102-112 (2018).
- 9 Ren, Z., Kim, J. C. & Lee, J. Transient cooling and heating effects in holey silicon-based lateral thermoelectric devices for hot spot thermal management. *IEEE Transactions on Components, Packaging and Manufacturing Technology* **11**, 1214-1222 (2021).
- 10 Mao, J., Chen, H., Jia, H. & Qian, X. The transient behavior of Peltier junctions pulsed with supercooling. *Journal of Applied Physics* **112** (2012).
- 11 Snyder, G. J., Fleurial, J.-P., Caillat, T., Yang, R. & Chen, G. Supercooling of Peltier cooler using a current pulse. *Journal of Applied Physics* **92**, 1564-1569 (2002).
<https://doi.org:10.1063/1.1489713>

- 12 Li, W.-K., Chang, J.-H., Amani, M., Yang, T.-F. & Yan, W.-M. Experimental study on transient supercooling of two-stage thermoelectric cooler. *Case Studies in Thermal Engineering* **14**, 100509 (2019).
- 13 Yang, R., Chen, G., Kumar, A. R., Snyder, G. J. & Fleurial, J.-P. Transient cooling of thermoelectric coolers and its applications for microdevices. *Energy Conversion and Management* **46**, 1407-1421 (2005).
- 14 Hamann, H. F. *et al.* Hotspot-Limited Microprocessors: Direct Temperature and Power Distribution Measurements. *IEEE Journal of Solid-State Circuits* **42**, 56-65 (2007).
<https://doi.org/10.1109/JSSC.2006.885064>
- 15 Yang, R., Chen, G., Snyder, G. J. & Fleurial, J.-P. Geometric effects on the transient cooling of thermoelectric coolers. *MRS Online Proceedings Library* **691**, 1-6 (2001).
- 16 Chakraborty, A. & Ng, K. C. Thermodynamic formulation of temperature–entropy diagram for the transient operation of a pulsed thermoelectric cooler. *International Journal of Heat and Mass Transfer* **49**, 1845-1850 (2006).
- 17 Thonhauser, T., Mahan, G., Zikatanov, L. & Roe, J. Improved supercooling in transient thermoelectrics. *Applied Physics Letters* **85**, 3247-3249 (2004).

REVIEWER COMMENTS

Reviewer #1 (Remarks to the Author):

Although the paper is much improved by authors in this version, I still have several questions to be answered.

1. Author added the thermoelectric parameters, and Seebeck coefficient and electrical resistivity of the p-doped silicon are adopted as $350 \mu\text{V K}^{-1}$ and $1 \times 10^3 \text{ S/cm}$. But, these values are too high for p-type bulk silicon, which are hardly realized in the doping level of 10^{21} cm^{-3} . (Nature volume 451, pages 163–167 (2008), Adv. Funct. Mater. 2009, 19, 2445–2452, and some other published data.)

2. Author mentioned that "Second and higher harmonic temperature profiles are excluded in the process of solving the heat equation, as they do not have a significant impact on the solution, as validated by our experiments.".... However, the phase-locked amplifier in the FDTR experiment has filtered out the higher-order harmonic signal, resulting in the signal detected by the experiment does not contain the information of the temperature change of the higher-order harmonic term, so the experimental verification is untenable. Can it be shown by theoretical calculation or simulation that the influence of second and higher harmonics on temperature can be ignored?

3. I agree that $T_0\omega$ will be affected by DC offset. But basically, because the laser energy is always positive, but the frequency matched AC current is half positive and half negative, although the frequency match could offset the $T_1\omega$ by the half negative part of the AC current. The other half part will still lead to a temperature rise, which results in an increase of steady temperature baseline, i.e. $T_0\omega$. Theoretical and experimental values of $T_0\omega$ are not given in this paper. If the application scenario is to create a stable thermal environment, the specific value of $T_0\omega$ does not need to be considered. However, when the application scenario is to use transient active cooling to reduce the peak temperature, the study of $T_0\omega$ or T_{max} is very important. Is it possible to measure $T_0\omega$ or T_{max} experimentally and compare them with theoretical values?

Reply to Referee and Revisions Made

Nature Communications Manuscript #: NCOMMS-23-39295A

Title: “Active Cooling for Transient Hot Spots in Microprocessors”

Authors: Yihan Liu, Hao-Yuan Cheng, Jonathan A. Malen, Feng Xiong

We appreciate the comments and feedback regarding our manuscript referenced above. We carefully considered all the comments and addressed them with point-by-point responses below, as well as through changes to the text of the manuscript. These comments requested additional acknowledgement of the steady-state temperature rise and the effect of higher harmonics during the process of first harmonic temperature active cooling. The reviewer comments are reproduced verbatim in *blue italics* below, followed by our responses and revisions made accordingly. The changes in the re-submitted manuscript itself have been **highlighted**. We are also providing a clean version of the re-submitted manuscript where the highlights are removed.

Reviewer #1

Although the paper is much improved by authors in this version, I still have several questions to be answered.

1. Author added the thermoelectric parameters, and Seebeck coefficient and electrical resistivity of the p-doped silicon are adopted as $350 \mu\text{V K}^{-1}$ and $1 \times 10^{-3} \text{ S/cm}$. But, these values are too high for p-type bulk silicon, which are hardly realized in the doping level of 10^{21} cm^{-3} .

(Nature volume 451, pages 163–167 (2008), Adv. Funct. Mater. 2009, 19, 2445–2452, and some other published data.)

We thank the reviewer for the thoughtful consideration. The value of electrical resistivity, $1 \times 10^{-3} \Omega\text{-cm}$, for the bulk p-Si was provided by our wafer supplier. We experimentally verified it by 4-probe measurement and the measured values, averaging $0.00105 \Omega \text{ cm}$, are plotted in Fig. R1a. Based on the given electrical resistivity value, we estimate that the doping level of the p-Si we used is between $1 \times 10^{20} \text{ cm}^{-3}$ and $1 \times 10^{21} \text{ cm}^{-3}$, referencing the Fig. 7 in Ref. [1]. The Seebeck coefficient of $350 \mu\text{V}\cdot\text{K}^{-1}$ was obtained by experimentally measuring the planar bulk p-Si wafer and the data are plotted in Fig. R1b as a function of temperature. We further compared the

measured Seebeck coefficient with other published data². A Seebeck coefficient of $375 \mu\text{V}\cdot\text{K}^{-1}$ was reported for a bulk p-Si sample with a doping level of $8.1\times 10^{19} \text{ cm}^{-3}$. Therefore, we consider our measured value of $350 \mu\text{V}\cdot\text{K}^{-1}$ to be reasonable for a p-Si sample with a doping level between $1\times 10^{20} \text{ cm}^{-3}$ and $1\times 10^{21} \text{ cm}^{-3}$.

Figure R1. Measurement of p-Si's electrical resistivity and Seebeck coefficient. **a** 50 4-probe measurements on a planar p-Si wafer are implemented to obtain the electrical resistivity of the bulk p-Si. The average resistivity is $0.00105 \Omega \cdot \text{cm}$. **b** Dependence of Seebeck coefficient is measured on a planar p-Si sample. A Seebeck coefficient of $350\pm 10 \mu\text{V K}^{-1}$ is obtained at 300 K.

We carefully reviewed the two reference papers provided by the reviewer^{3,4}. The first paper reports the Seebeck coefficient and electrical resistivity of heavily p-doped Si nanowires, as well as some thermoelectric properties of n-type bulk Si, but it does not mention the doping level of the p-type Si nanowire and the thermoelectric properties of heavily p-doped bulk Si. Similarly, the second paper only includes thermoelectric properties of n-type nano-bulk Si and intrinsic bulk Si without including data for heavily doped p-type bulk Si. Therefore, the data presented in these two papers do not contradict to our measured electrical resistivity or Seebeck coefficient of the p-Si we used.

However, the doping level value of p-Si, which we estimated based on the resistivity measurement, is not critical to our conclusions. Therefore, we have removed the doping level

value from the manuscript, retaining only the experimentally verified electrical resistivity and Seebeck coefficient.

We edit the following text in section “Method”, line 354-356:

Substrates consist of thermal SiO₂ of 100 nm thickness, grown on highly p-doped silicon (the measured values of Seebeck coefficient and electrical resistivity of the p-doped silicon are 350 μV K⁻¹ and 1.05×10⁻³ Ω cm.).

2. Author mentioned that "Second and higher harmonic temperature profiles are excluded in the process of solving the heat equation, as they do not have a significant impact on the solution, as validated by our experiments."... However, the phase-locked amplifier in the FDTR experiment has filtered out the higher-order harmonic signal, resulting in the signal detected by the experiment does not contain the information of the temperature change of the higher-order harmonic term, so the experimental verification is untenable. Can it be shown by theoretical calculation or simulation that the influence of second and higher harmonics on temperature can be ignored?

Due to the second order harmonic component of Joule heating from the first harmonic current, $J_{1\omega}^2 \rho$, a second harmonic temperature response that grows with applied current will emerge.

To quantify this, we manually set the demodulator frequency of the lock-in amplifier (HF2LI, Zurich Instruments) to measure the amplitude of higher harmonic components in the transient temperature response. When the first harmonic temperature was experimentally actively cancelled (i.e., when $T_{1\omega}$ is close to 0), we did not observe significant higher order harmonic signals. Data for the first harmonic and the four higher harmonics, shown in Fig. R2, show that the 2nd harmonic is largest, and at most 15% of the first harmonic amplitude before cancellation.

Figure R2. Higher harmonic temperature response when the first harmonic temperature is actively cancelled. When the first harmonic temperature is actively cooled down from 2 K to 0.1 K with applied AC current of 70mA, measured 2nd harmonic temperature component is only 0.3 K. Applied laser power is 658 μ W in modulation frequency of 10 kHz. Higher harmonic noises resulting from the EOM and function generator are not subtracted.

Additionally, the thermoelectric heat flux generated from the 2nd harmonic temperature component and the 1st harmonic current, $SJ_{1\omega}T_{2\omega}$, will impact the first harmonic temperature. A larger applied current will result in a larger amplitude this thermoelectric heat flux will be due to higher $J_{1\omega}$ and $T_{2\omega}$. However, we found in our experiments that the transient temperature modulation capability, T_{δ} , did not change with the applied current. This means that in Fig. 3b's upper panel, $T_{1\omega}$ decreases linearly with the increase in $J_{1\omega}$ (T_{δ} 's amplitude does not change), and in Fig. 3c, the measured change in $T_{1\omega}$ is linear as current increases in the polar coordinate axes (T_{δ} 's phase does not change). Therefore, the data shown in Fig. 3 indirectly supports the validity of our simplification in ignoring higher order harmonic temperature response in modelling study.

We add this discussion in Supplementary Information, line 230-241:

Supplementary Note 8: Discussion on the Simplification of Neglecting Higher Order Transient Temperature Components in Model Study

In the model study, we neglect higher order harmonic temperature and its influence on thermoelectric effect to obtain a closed-form result for the 1st harmonic temperature response. In reality the 2nd harmonic from Joule heating will generate a 2nd harmonic temperature response, $T_{2\omega}$, that grows with applied current. This $T_{2\omega}$ will also affect $T_{1\omega}$ through the thermoelectric heat flux $SJ_{1\omega}T_{2\omega}$. Nevertheless, Fig. 3b and 3c show that the transient temperature modulation capability, T_{δ} , is independent of the amplitude of the AC current. In Fig. 3b's upper panel, the linear reduction of $T_{1\omega}$ indicates that T_{δ} 's amplitude is independent AC current. In Fig. 3c, the linear trajectory of $T_{1\omega}$ in polar coordinates indicates that the phase of T_{δ} has not changed. Therefore, data shown in Fig. 3 indicate that higher order harmonic temperatures are small and do not significantly inhibit the first harmonic temperature active cooling process.

3.I agree that $T_{0\omega}$ will be affected by DC offset. But basically, because the laser energy is always positive, but the frequency matched AC current is half positive and half negative, although the frequency match could offset the $T_{1\omega}$ by the half negative part of the AC current. The other half part will still be leading to a temperature rise, which results in an increase of steady temperature baseline, i.e. $T_{0\omega}$. Theoretical and experimental values of $T_{0\omega}$ are not given in this paper. If the application scenario is to create a stable thermal environment, the specific value of $T_{0\omega}$ does not need to be considered. However, when the application scenario is to use transient active cooling to reduce the peak temperature, the study of $T_{0\omega}$ or T_{max} is very important. Is it possible to measure $T_{0\omega}$ or T_{max} experimentally and compare them with theoretical values?

Based on the results of our model, the thermoelectric effect generated by the AC current only affects the first harmonic temperature. However, even without a DC offset, the steady-state component of Joule heating generated by the AC current will cause $T_{0\omega}$ to increase.

In our manuscript, we have provided a theoretical estimate of the change in $T_{0\omega}$, $\Delta T_{0\omega} = \frac{|J_{1\omega}|^2 \rho_{Si} (z_2 - z_1)^2}{4\kappa_{Si}}$, and depicted it in Fig. 3b (gray solid line). With this estimate, we can calculate the reduction in T_{max} as $\Delta T_{max} = T_{1\omega 0} - T_{1\omega} - \Delta T_{0\omega}$ (as depicted in Fig. R3) and this value can be extracted from the data of $T_{1\omega}$ and the estimate of $\Delta T_{0\omega}$ we provided in Fig. 3b. However, since we cannot directly measure the steady-state temperature rise using a lock-in amplifier, we

cannot provide experimental data to compare with this theoretical estimation of $\Delta T_{0\omega}$ and ΔT_{\max} . Since our main conclusions emphasize cancellation of the 1st harmonic for stabilization, we feel that additional measurements of $T_{0\omega}$ are outside the scope of this manuscript.

Figure R3. Sketch of transient temperature before and after applying transient active cooling. Black dash line represents the steady-state temperature. The steady-state component of the AC's Joule heating would induce an increase in $T_{0\omega}$. Red dash line depicts the max temperature level before and after applying transient active cooling. The absolute value of T_{\max} reduction is the net result from the reduction of the first harmonic temperature and the increase in steady-state temperature.

To emphasize the effect of steady-state temperature increase on peak temperature's absolute value, we include the following text in the manuscript, section "Results and discussion", line 245-246:

The transient peak temperature is a balance of the two effects that depends on the magnitude of $J_{1\omega}$ and the operating frequency.

Reference

- 1 Masetti, G., Severi, M. & Solmi, S. Modeling of carrier mobility against carrier concentration in arsenic-, phosphorus-, and boron-doped silicon. *IEEE Transactions on Electron Devices* **30**, 764-769 (1983).
- 2 Stranz, A., Kähler, J., Waag, A. & Peiner, E. Thermoelectric Properties of High-Doped Silicon from Room Temperature to 900 K. *Journal of Electronic Materials* **42**, 2381-2387 (2013). <https://doi.org/10.1007/s11664-013-2508-0>
- 3 Hochbaum, A. I. *et al.* Enhanced thermoelectric performance of rough silicon nanowires. *Nature* **451**, 163-167 (2008). <https://doi.org/10.1038/nature06381>
- 4 Bux, S. K. *et al.* Nanostructured Bulk Silicon as an Effective Thermoelectric Material. *Advanced Functional Materials* **19**, 2445-2452 (2009). <https://doi.org/https://doi.org/10.1002/adfm.200900250>

REVIEWERS' COMMENTS

Reviewer #1 (Remarks to the Author):

OK, i have no question about this work.